DOI: 10.1038/s41467-018-04708-5　　OPEN

# Interplay of the two ancient metabolites auxin and MEcPP regulates adaptive growth

Jishan Jiang[1], Cecilia Rodriguez-Furlan [1], Jin-Zheng Wang[1], Amancio de Souza[1], Haiyan Ke[1], Taras Pasternak[2], Hanna Lasok[2], Franck A. Ditengou [2], Klaus Palme[2] & Katayoon Dehesh[1]

The ancient morphoregulatory hormone auxin dynamically realigns dedicated cellular processes that shape plant growth under prevailing environmental conditions. However, the nature of the stress-responsive signal altering auxin homeostasis remains elusive. Here we establish that the evolutionarily conserved plastidial retrograde signaling metabolite methylerythritol cyclodiphosphate (MEcPP) controls adaptive growth by dual transcriptional and post-translational regulatory inputs that modulate auxin levels and distribution patterns in response to stress. We demonstrate that in vivo accumulation or exogenous application of MEcPP alters the expression of two auxin reporters, DR5:GFP and DII-VENUS, and reduces the abundance of the auxin-efflux carrier PIN-FORMED1 (PIN1) at the plasma membrane. However, pharmacological intervention with clathrin-mediated endocytosis blocks the PIN1 reduction. This study provides insight into the interplay between these two indispensable signaling metabolites by establishing the mode of MEcPP action in altering auxin homeostasis, and as such, positioning plastidial function as the primary driver of adaptive growth.

[1] Department of Botany and Plant Sciences and Institute of Integrative Genome Biology, University of California, Riverside, CA 92506, USA. [2] University of Freiburg, Faculty of Biology; BIOSS Centre for Biological Signaling Studies and ZBSA Centre for Biosystems Studies, Schänzlestr. 1 79104 Freiburg, Germany. These authors contributed equally: Jishan Jiang, Cecilia Rodriguez-Furlan. Correspondence and requests for materials should be addressed to K.D. (email: kdehesh@ucr.edu)

Plant adaptive growth is a central strategy evolved through integration and coordination of a complex composite of signaling pathways to cope with inevitable environmental challenges. Auxin [indole-3-acetic acid (IAA)] is an indispensable hormone functional as a morphoregulatory informational output integrating developmental and environmental cues into a complex regulatory network enabling optimal architectural modifications both in plantae and eubacterial kingdoms[1–5].

It is well established that sculpting and adjusting the duration of adaptive responses are in part regulated by the dynamic alterations in IAA levels and distribution patterns[3,6–10]. The flavin mono-oxygenases YUCCA (YUC) family of enzymes are the rate limiting step in IAA biosynthesis in plants[11–13]. The analyses of various *yuc* mutant combinations demonstrated the critical role of local auxin biosynthesis in seedling growth and developmental processes[14,15]. The dynamics of auxin response is often monitored using the synthetic auxin-responsive promoter *DR5* or Aux/IAA auxin-interaction domain DII fused to a reporter, enabling the visualization of the spatial pattern of auxin response and thus auxin gradients[16–19]. The establishment of auxin gradients supported by cellular efflux requires the functional network of auxin-efflux carrier family of PIN-FORMED (PIN) proteins comprised of eight members in Arabidopsis[20]. The founding family member is PIN1, initially identified by characterization of the *pin-formed1* mutant in Arabidopsis, is a recycling membrane protein localized on the basal side of cells in the vascular tissue[21–23]. The delivery of newly synthesized PIN1 to plasma membrane requires ARF guanine-nucleotide exchange factors (ARF-GEFs), BIG1 through BIG4, while the abundance of PIN1 at the plasma membrane, and the consequential auxin distribution is regulated by clathrin-mediated endocytosis[23–26].

Beyond being a core regulator of an array of plant developmental processes including growth and architecture, auxin is also an instrumental hormone in tailoring responses to abiotic and biotic stimuli[27]. As such, the interplay between environmental inputs and the control on auxin levels and distribution patterns provide the plasticity required for plant's survival, as reflected in stress-mediated morphogenic responses by reduced cell division and elongation[28]. In fact, auxin homeostasis refines plant responses to environmental signals by multiple mechanisms including modulation of reactive oxygen species (ROS) homeostasis, induction of ROS detoxification enzymes, and regulation of chloroplast protein import[29–32]. Reciprocally, auxin metabolism is regulated by plastidial enzymes such as the plastidial NADPH-thioredoxin reductase (NTRC). This enzyme is reported to be involved in regulation of auxin levels as evidenced by reduced auxin and hence retarded growth of *ntrc* mutant as compared to the control plants[33]. Collectively, these results have led to the assumption that plastidial stress-signaling metabolite(s) may play a role in maintaining auxin homeostasis and by extension regulation of plant adaptive responses. Specifically, a connection between auxin and plastid-to-nucleus (retrograde) signaling is suggested, and a role for auxin-based signaling as secondary components involved in the response cascades following a retrograde plastidial signal is assumed[34].

The stunted growth phenotype of the *ceh1* (constitutively expression HPL) mutant triggered our interest to explore the plausible connection between auxin and a stress-specific plastidial retrograde signaling metabolite in plant adaptive responses. Specifically, we questioned whether altered auxin homeostasis may contribute to the stunted phenotype of the *ceh1* mutant, and if so how auxin homeostasis might be altered in this plant. The focus on *ceh1* is because this mutant accumulates methylerythritol cyclodiphosphate (MEcPP), an essential bifunctional plastidial metabolite serving as a precursor of isoprenoids produced by the plastidial methylerythritol phosphate (MEP)

pathway, as well as a stress-specific retrograde signal communicating environmental perturbations sensed by plastid-to-nucleus[35–39]. The MEP pathway is evolutionarily conserved, and MEcPP is produced not only by plantae but also by the non-photosynthetic "apicoplast" plastids of parasites such as the malarial parasite, and by eubacteria[36,39]. Intriguingly, stress-mediated accumulation of MEcPP in bacterial culture suggests the ancient nature and functional conservation of this metabolite beyond plantae[36,39–41].

Here, using constitutive and inducible MEcPP accumulating lines, in concert with pharmacological interference with the flux through MEP pathway, and exogenous treatment of plants with MEcPP, enabled us to establish the specifc and the key role of this stress-specific plastidial retrograde signaling metabolite in modulating growth by reducing the abundance of auxin and its transporter PIN1 via dual transcriptional and post-translational regulatory inputs.

## Results

**MEcPP-dependent reduction of IAA and PIN1 abundance.** The stunted growth of the *ceh1* prompted us to examine whether this compromised phenotype is caused by constitutively high MEcPP or by increased levels of salicylic acid (SA) in the mutant background[39]. These studies were further warranted by the established SA-mediated suppression of the auxin signaling[42]. Thus, we analyzed the hypocotyl length of seedlings in four genotypes P, *ceh1*, the SA deficient lines *eds16*, and *ceh1 eds16* double mutant[39]. Similarly retarded growth of *ceh1* and *ceh1eds16* seedlings unequivocally demonstrate an SA-independent, but MEcPP-dependent inhibition of hypocotyl growth in *ceh1* (Supplementary Fig. 1a–c).

Next, we examined the potential role of IAA in the stunted hypocotyl phenotype of *ceh1* seedlings[39]. Measurements of IAA established the reduced levels of this growth hormone in *ceh1* compared to lines originally employed in the genetic screen that identified *ceh1*, and thus designated as Parent (P) plants[39] (Fig. 1a). Notable recovery of the *ceh1* short hypocotyl phenotype by exogenous application of IAA reaffirmed the involvement of this growth hormone (Fig. 1b, c). Application of various concertation of auxin, however, did not result in an increased root length in *ceh1*. Nevertheless, low concentration of auxin (1 μM), resulted in a less severely reduced growth of roots in *ceh1* compared to P plants (Supplementary Fig. 2). Collectively, these results suggest effectiveness of IAA in modulating both hypocotyl and root growth in *ceh1* albeit at different degrees.

The established role of MEcPP as a key dynamic orchestrator of transcriptional network[43] led us to question whether the reduced IAA level is in part due to alteration in transcript levels of YUCCA (YUC) genes encoding a family of enzymes catalyzing the rate limiting step in IAA biosynthesis[11–13]. The expression level analyses of YUC family members (YUC1 through YUC10) show reduced YUC3 and YUC5 transcripts in *ceh1* compared to the P (Fig. 1d & Supplementary Fig. 3). This selectivity supports the earlier notion of an overlapping function among a subset of the YUCs[15], and further promotes the tailored functionality of individual YUCs in response to specific stress signal(s).

Next, we monitored dynamics of auxin response using the synthetic auxin-responsive promoter *DR5*:GFP. The *ceh1* mutant expressing *DR5*:GFP showed a ~70% reduction in *DR5* expression as compared to P (Fig. 1e & Supplementary Fig. 4a).

To assess the contribution of auxin transport to the decreased *DR5* promoter-driven GFP signal in the *ceh1* mutant, we examined the transcript levels of the auxin-efflux carrier PIN-FORMED1 (PIN1), a recycling membrane protein localized on the basal side of cells in the vascular tissue[23]. Expression

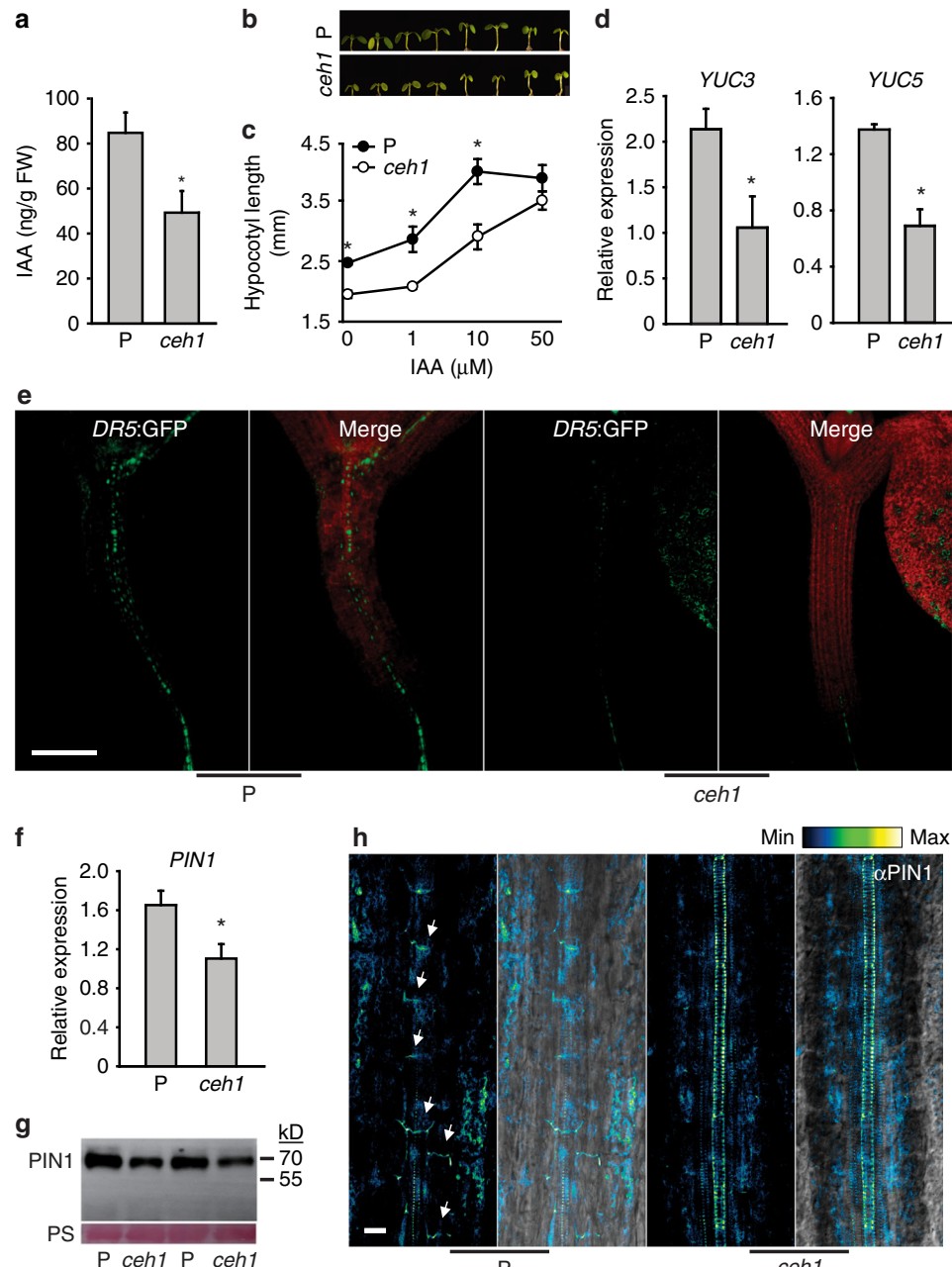

**Fig. 1** Auxin and PIN1 are decreased in *ceh1*. **a** Reduced IAA content in *ceh1*. Data are expressed as mean ± SD, n ≥ 50. **b** Representative images of hypocotyl length in control (P) and *ceh1* seedlings grown in the presence of different IAA concentrations. **c** Quantitative measurements of hypocotyl length from panel (**b**). Data are expressed as mean ± SD; n ≥ 45. **d** Relative mRNA levels of *YUC3* and *5* in *ceh1*, expressed as mean ± SD of three biological and three technical repeats each. **e** Representative images of DR5-GFP (green) in 7-day old hypocotyls of P and *ceh1*, chloroplast fluorescence (red) and the merged images. The experiments were performed three times, each with 10 biological replicates. Scale bar: 100 μm. Transcript (**f**) and protein (**g**) levels of PIN1 in P and *ceh1* plants. Ponceau S (PS) staining displays equal protein loading. **h** Representative PIN1 immunolocalization in hypocotyls of P and *ceh1* using αPIN1 antibody (left) and merged image with bright field (right) depict reduced PIN1 levels in *ceh1*. Images are from two independent experiments, each with 10 biological replicates. Scale bar: 20 μm. The color-coded bar displays PIN1 fluorescence intensity. Asterisks indicate significant differences as determined by a two-tailed Student's *t* tests with a significance of *P* < 0.05

analyses of *PIN1* in *ceh1*, followed by extended analyses in the four aforementioned genotypes demonstrated a reduction in *PIN1* transcript levels exclusively in high MEcPP containing lines (Fig. 1f and Supplementary Fig. 4b). Furthermore, the combined approaches of Western blot and immunolocalization analyses confirmed a ~40% reduction in PIN1 protein levels in *ceh1* compared with P (Fig. 1g-h & Supplementary Fig. 4c).

The physiological ramification of the altered auxin homeostasis in *ceh1*, especially in light of the reduced susceptibility of the mutant to auxin inhibition of root growth (Supplementary Fig. 2), led us to compare the root cell cycle in mutant versus the P plants. To examine the cell-cycle progression we employed EdU, a thymidine analog that incorporates only in DNA during replication, enabling a direct and quantitative measure at

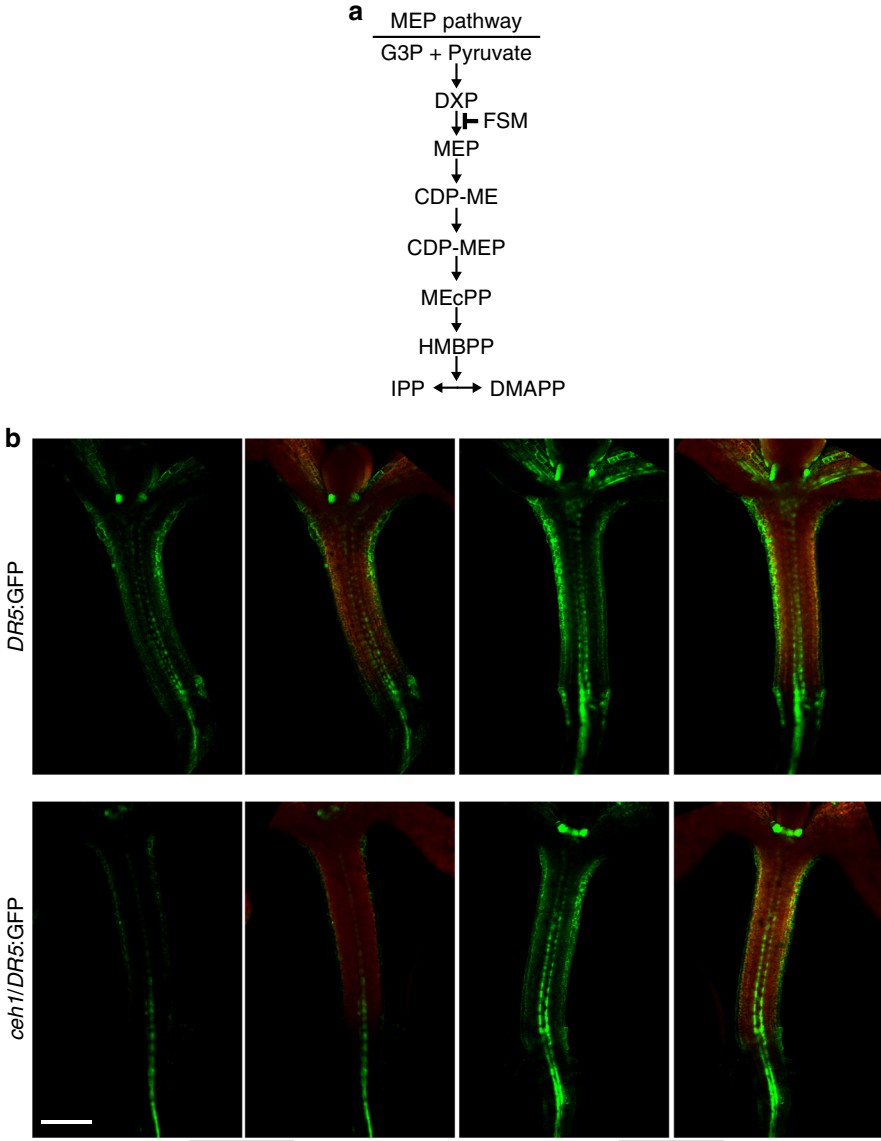

**Fig. 2** MEP pathway inhibition by fosmidomycin recovers *DR5*:GFP signal in *ceh1* seedlings. **a** Schematic representation of MEP pathway noting the site of fosmidomycin (FSM) inhibitory action. **b** Representative images of *DR5*:GFP (green) in 7-day-old hypocotyls of P (*DR5*:GFP) and *ceh1*, chloroplast fluorescence (red) and the merged images. The experiments were performed three times, each with 10 biological replicates. Scale bar: 100 μm

single-cell resolution at the root tips by the intrinsic root coordinate system (iRoCS) method[44,45]. The analyses of cell-cycle events in the root apical meristem (RAM) revealed reduced DNA replication and by extension cell division events in the *ceh1* compared with the P (Supplementary Fig. 5a). Accordingly, the reduced cortex cell number in RAM resulted in shorter meristem size of *ceh1* roots (Supplementary Fig. 5b).

To further examine the potential role of MEcPP in *ceh1* mutant in reducing growth and altering auxin homeostasis as measured by the *DR5*:GFP signal, we employed a pharmacological approach using fosmidomycin (FSM), a MEP-pathway inhibitor (Fig. 2a). This inhibitor arrests the flux through the pathway and abolishes MEcPP-mediated formation of otherwise stress-induced subcellular structures known as ER bodies in the *ceh1* mutant[46,47]. The analyses of *DR5*:GFP expression in 7-day-old seedlings grown in the presence of the inhibitor clearly show that while the FSM treatment did not modulate *DR5* expression in P seedlings, it did recover the low *DR5*:GFP expression in *ceh1/DR5*:GFP seedlings to levels similar to that in P plants (Fig. 2b & Supplementary Fig. 6).

Collectively, these observations confirmed MEcPP-mediated alteration of auxin homeostasis as a key mechanism underpinning growth retardation in the *ceh1* mutant.

**Induction of MEcPP reduces the abundance of auxin and PIN1**. To further explore the MEcPP potential function in altering auxin levels and PIN1 protein abundance, we generated dexamethasone (DEX) inducible RNAi lines of hydroxyl methyl butenyl diphosphate synthase (*HDS*), encoding the enzyme catalyzing the conversion of MEcPP to hydroxymethylbutenyl diphosphate[39]. Metabolic analyses showed increasing MEcPP levels at 48 and 72 h post DEX induction in concert with the reduction of auxin levels in *HDS RNAi* (*HDSi*) lines, as compared to P plants (Fig. 3a–b). In addition, Western blot analyses clearly illustrated decreasing PIN1 protein abundance in *HDSi* lines at 48 and 72 h post induction relative to levels examined in seedling analyzed immediately after induction (0 time) (Fig. 3c).

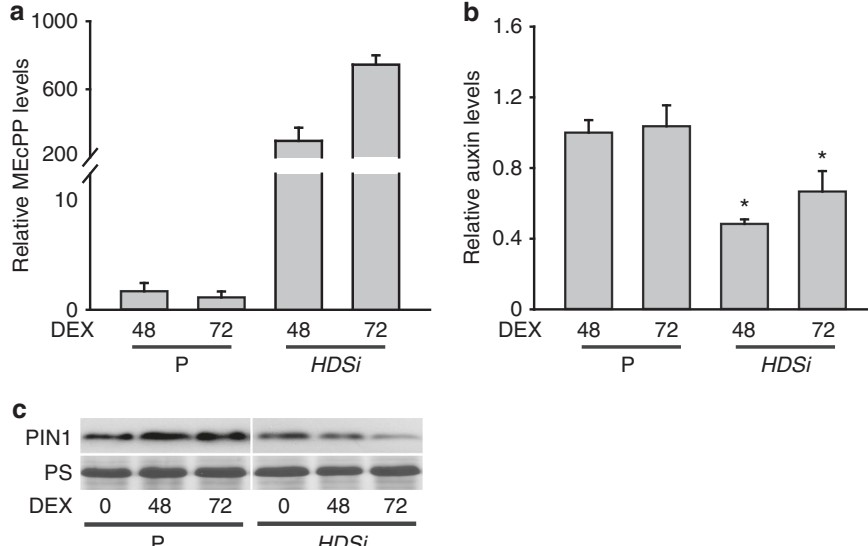

**Fig. 3** Accumulation of MEcPP in inducible *HDSi* correlates with reduction of auxin and PIN1 abundance. **a** Levels of MEcPP in P and *HDSi* at 48 h and 72 h post DEX induction relative to un-induced seedlings. **b** Levels of auxin in P and *HDSi* at 48 h and 72 h post DEX induction relative to un-induced seedlings. **c** Protein levels of PIN1 in P and *HDSi* before and at 48 h and 72 h post DEX induction. Ponceau S (PS) staining displays equal protein loading

Collectively, the data from the inducible *HDSi* lines display a clear correlation between accumulation of MEcPP and a reduction in IAA and PIN1 abundance, thus further substantiate the finding obtained from constitutively producing MEcPP mutant plant (*ceh1*) and corroborate with FSM treatment results.

**Exogenous MEcPP reduces auxin levels and PIN1 abundance.** To test the functional specificity of MEcPP signaling in altering auxin homeostasis, we visualized auxin distribution in response to exogenous application of MEcPP in two well-established independent maker lines *DR5*:GFP and DII-VENUS[16–19]. We specifically examined the *DR5*:GFP signal in hypocotyls of Mock and MEcPP treated seedlings and established reduced GFP signal post MEcPP treatment (Fig. 4a & Supplementary Fig. 7a). As expected and in contrast, the DII-VENUS signal in roots of MEcPP treated seedlings was enhanced as compared to the mock treated seedlings (Fig. 4b). Collectively, these results illustrate MEcPP-mediated reduction of auxin concentration in hypocotyls and roots of plants exogenously treated with the metabolite.

Furthermore, we monitored PIN1 abundance in response to exogenously applied MEcPP by examining PIN1-immuno-signal intensity in hypocotyls in conjunction with PIN1-GFP signal in the roots. The analyses show a rapid reduction in PIN1-immuno-signal intensity and PIN1:GFP signal at 10 min post MEcPP treatment, followed by further reductions at the later time point (Fig. 4c-d). These results are in agreement with the reduced *DR5*:GFP and PIN1 abundance in the inducible *HDSi* lines, and constitutively high MEcPP containing *ceh1* mutant plants (Supplementary Fig. 4a and 7a). The one inconsistency between these data set is the reduced expression of *PIN1* in the *ceh1* mutant as opposed to MEcPP-treated plants (Fig. 1f and Supplementary Fig. 7b). This suggests a MEcPP-mediated immediate and specific response at the PIN1 protein levels.

Next, we questioned whether the response to MEcPP action is specific to PIN1 or nonspecific extending to other plasma membrane proteins, such as PIN2 and PIN3, members of the PIN auxin-efflux carrier family and an unrelated plasma membrane marker, NPSN12[48,49]. Absence of any visible alteration in the signal intensity of PIN3 and NPSN12 proteins in hypocotyl following MEcPP application established specificity of PIN1 response (Supplementary Fig. 7c, d). Furthermore, the

result led to the conclusion that the decrease in the PIN3 signal intensity in the *ceh1* as compared to the P (Supplementary Fig. 7e) may be the consequence of general stress or due to the sustained decrease in auxin levels[50]. The modest reduction in PIN2:GFP signal in response to MEcPP application in roots however, suggest responsiveness of PIN2 to MEcPP albeit at a much lower magnitude than that of PIN1(Supplementary Fig. 8a). The YFP signal derived from the PIN unrelated plasma membrane protein NPSN12:YFP (Supplementary Fig. 8b) remains unaltered in responses to MEcPP treatment, thus further supporting targeted action of MEcPP.

**High light alters IAA homeostasis.** To assess the physiological relevance of MEcPP-mediated alteration of IAA homeostasis, we treated plants with high light (HL), a stress known to increase the MEcPP levels rapidly and transiently[39]. We specifically examined *DR5*:GFP distribution in conjunction with PIN1-immuno-signal intensity of P plants before and at various intervals of HL exposure (30, 60, and 90 min). These results demonstrate reduced *DR5*:GFP distribution and PIN1-immuno-density signal intensities at all the time points of HL treatment (Fig. 5a, b and Supplementary Fig. 9a). It is of note that the reduction in the PIN1-immuno-density signal is not supported by measurable changes in expression levels of *PIN1* in response to increasing MEcPP levels by HL (Supplementary Fig. 9b-c). Concordance of the data with the aforementioned result (Supplementary Fig. 7b) and discordance with reduced expression of PIN1 in constitutively high MEcPP containing *ceh1* (Fig. 1f), differentiates between the consequences of general stresses versus the MEcPP-specific responses, and further alludes to post-transcriptional regulatory action of MEcPP in reducing PIN1 abundance. The transient nature of the response is evident from the recovery of both *DR5*:GFP abundance and the PIN1-immuno-density signal coupled with the reduced MEcPP levels to the basal levels at 24 h post HL treatment (Fig. 5c, d and Supplementary Fig. 9d-f).

**High light enhances clathrin-mediated endocytosis of PIN1.** The specificity of MEcPP-mediated reduction of PIN1 abundance led to the question of whether the mode of MEcPP action is through internalization of PIN1 to the plasma membrane. Thus,

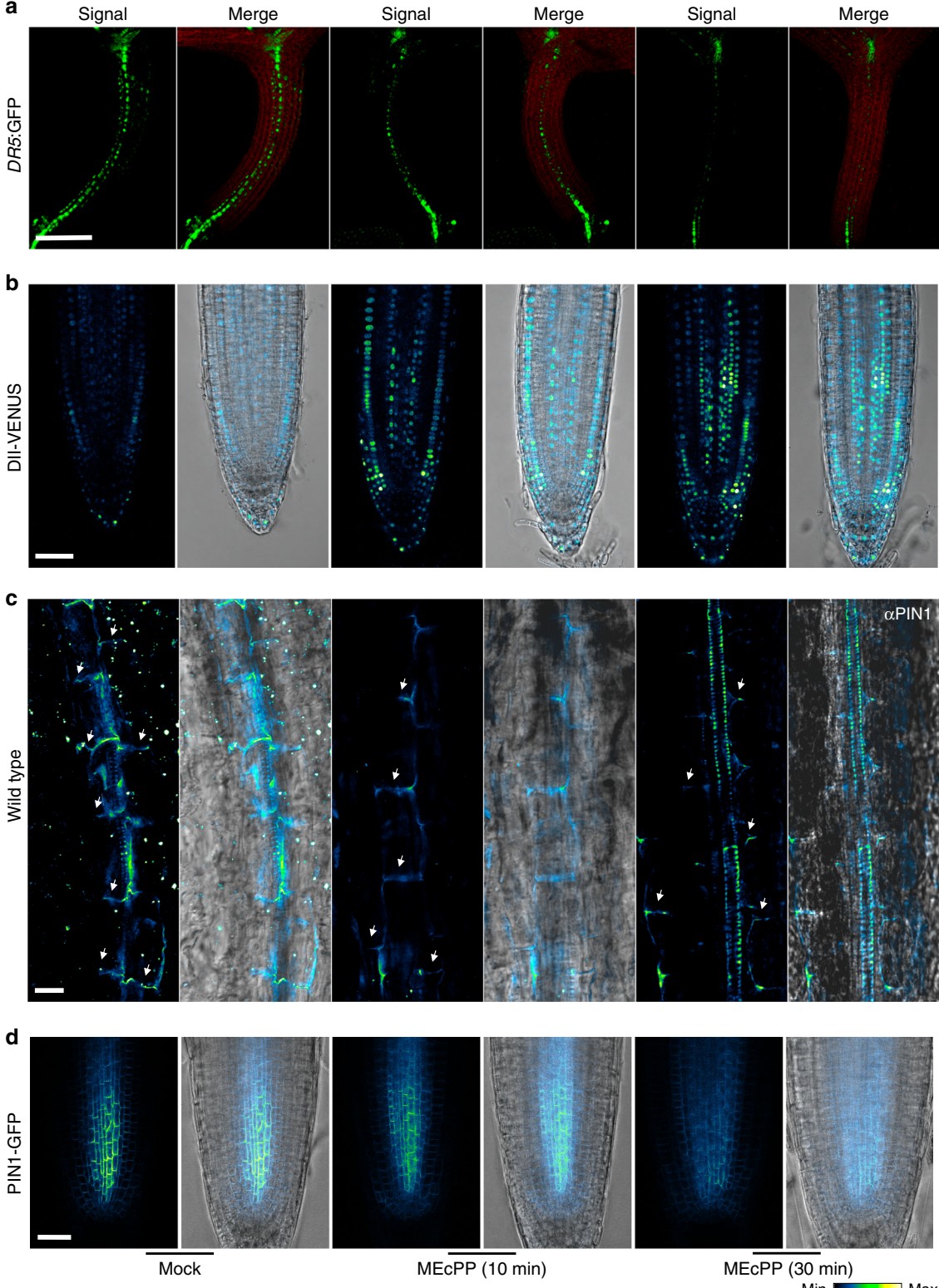

**Fig. 4** MEcPP reduces *DR5*:GFP and PIN1 abundance. **a**, **b** Representative images of **a** *DR5*:GFP and **b** DII-VENUS signal intensities in hypocotyls and roots of 7-day-old seedlings, before and at intervals after MEcPP application, respectively. *DR5*:GFP (green), chloroplast fluorescence (red), DII-VENUS (blue) from 10 independent experiments. **c**, **d** Representative images of **c** PIN1-imuno and **d** PIN1-GFP signal intensities in hypocotyls and roots of 7-day-old seedlings, before and at intervals after MEcPP application, respectively. **c** PIN1 immunolocalization using αPIN1 antibody (left) and merged image with bright field (right), from three independent experiments and $n \geq 8$ biological replicates each. The color-coded bar displays the PIN1 fluorescence intensity. Scale bars:100 μm (**a**) and 20 μm (**b**–**d**)

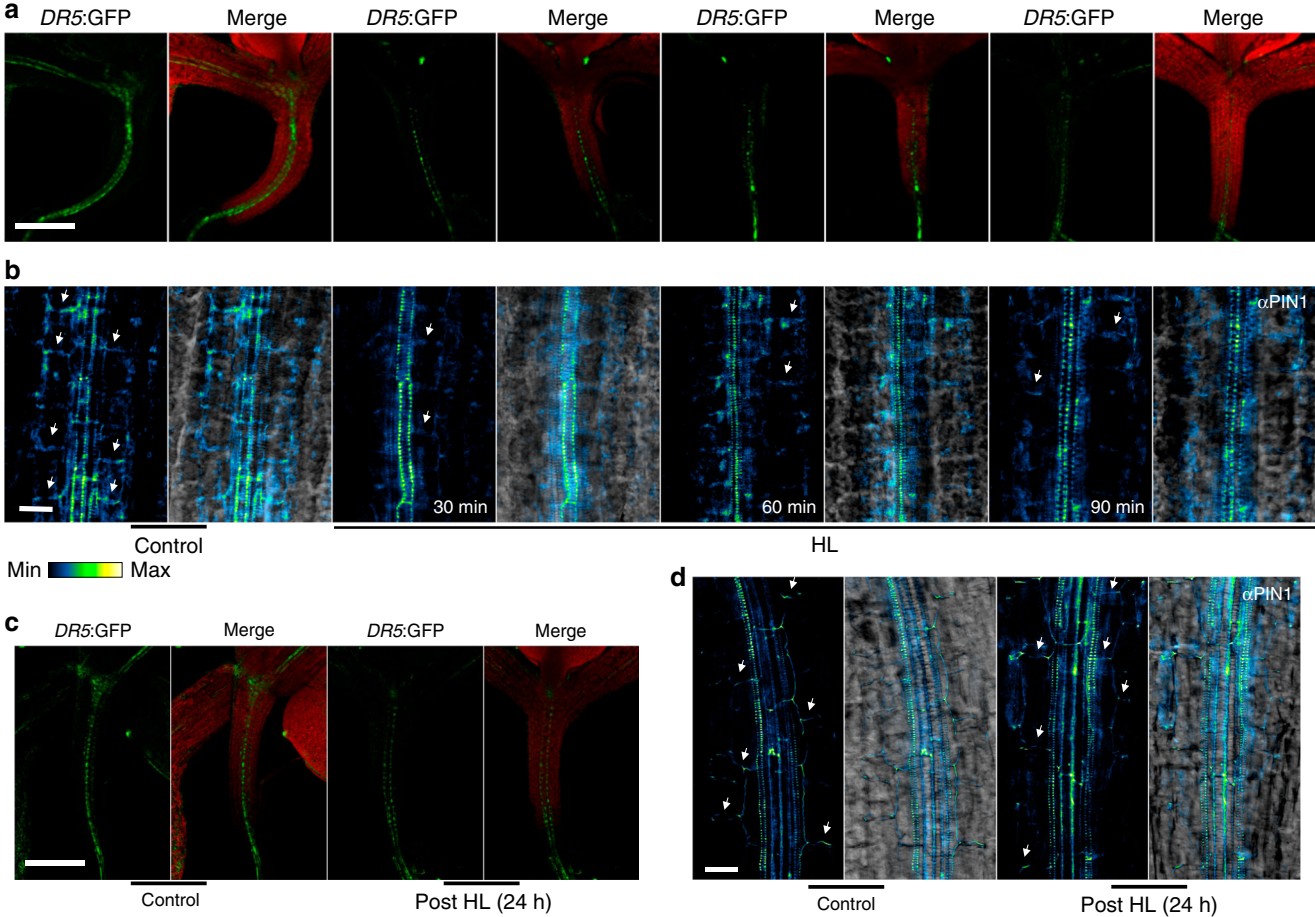

**Fig. 5** High light reduces *DR5*:GFP and PIN1 abundance. **a, b** Representative images of *DR5*:GFP and PIN1 abundance in 7-day hypocotyls of P seedlings after high light (HL) treatment. **a** *DR5*:GFP (green), chloroplast fluorescence (red) and merged images. **b** PIN1 immunolocalization in hypocotyls of P seedlings using αPIN1 antibody (left) and merged image with bright field (right). **c, d** Recovery of *DR5*:GFP and PIN1 at the PM at 24 h post HL treatment, respectively. The color-coded bar displays the PIN1 fluorescence intensity. All images are representatives of three independent experiments with $n \geq 8$ biological replicates each. Scale bars: 100 μm (**a**, **c**) and 20 μm (**b**, **d**)

we examined the potential role of BIG proteins required for delivery of newly synthesized and/or recycled PIN1 to the plasma membrane[51]. We specifically tested and compared PIN1-immuno-signal intensity in *big2,3,4* and *big1,2,4* mutant lines[26] before and after HL treatment (Fig. 6a). The notable reduction of PIN1 signal intensity in both mutant lines in response to HL excludes BIG proteins as the MEcPP path of action.

Next, we exploited pharmacological interference with clathrin-dependent endocytosis of PIN1 using tyrosine analog tyrphostin A23 (TyrA23), a well-established inhibitor of recruitment of endocytic cargo into the clathrin-mediated pathway, together with tyrphostin A51 (TyrA51) a close structural analog of TyrA23 routinely used as a negative control[25,52]. The unaltered PIN1-immuno-density signal in HL/TyrA23 treated as opposed to those of HL and HL/TyrA51 treated plants is a clear demonstration of clathrin-dependent endocytosis route of action (Fig. 6b). This strongly supports the notion that MEcPP-mediated signal(s) enable a precise control of auxin fluxes through post-transcriptional regulation of PIN1 abundance at the plasma membrane.

## Discussion

Plants exquisitely tune and align their growth to prevailing environmental conditions. Underpinning this adaptation is auxin, the morphoregulatory hormone that dynamically realigns dedicated cellular processes that shape growth under standard and

stress conditions. However, the nature of stress-responsive endogenous signaling molecule that regulates levels and distribution patterns of this hormone has remained elusive.

Here, the outcome of studies using constitutively and inducible MEcPP-producing lines in conjunction with pharmacological interference with the MEP pathway, and with exogenous application of MEcPP, established this stress-specific plastidial retrograde signal, MEcPP, as the upstream signal defining the optimal abundance of IAA and PIN1 via dual transcriptional and post-transcriptional regulatory inputs. Specifically, MEcPP accumulation in response to stress signals[39,47] reduces growth by altering IAA level. Indeed, the pharmacological hinderance of flux though the MEP pathway substantiates the role of MEcPP in modulation of auxin abundance, as examined by *DR5*:GFP signal. Our data further support the notion that reduction in auxin levels is in part through decreased levels of *YUC3* and *5* transcripts. It is of note that a previous report has clearly demonstrated that mutation in five *YUC* genes (*YUC3*, *YUC5*, *YUC7*, *YUC8* and *YUC9*) resulted mainly in retarded development of roots, and not hypocotyls of mutant seedlings[53]. Accordingly, we propose that MEcPP-mediated stunted *ceh1* hypocotyl growth is not exclusively due to reduced expression of *YUC* genes but it is also the result of reduced auxin transport.

The mode of MEcPP action in transcriptional suppression of *YUC* genes is yet to be determined, but the notion of integration of MEcPP into transcriptional networks and robust alteration of

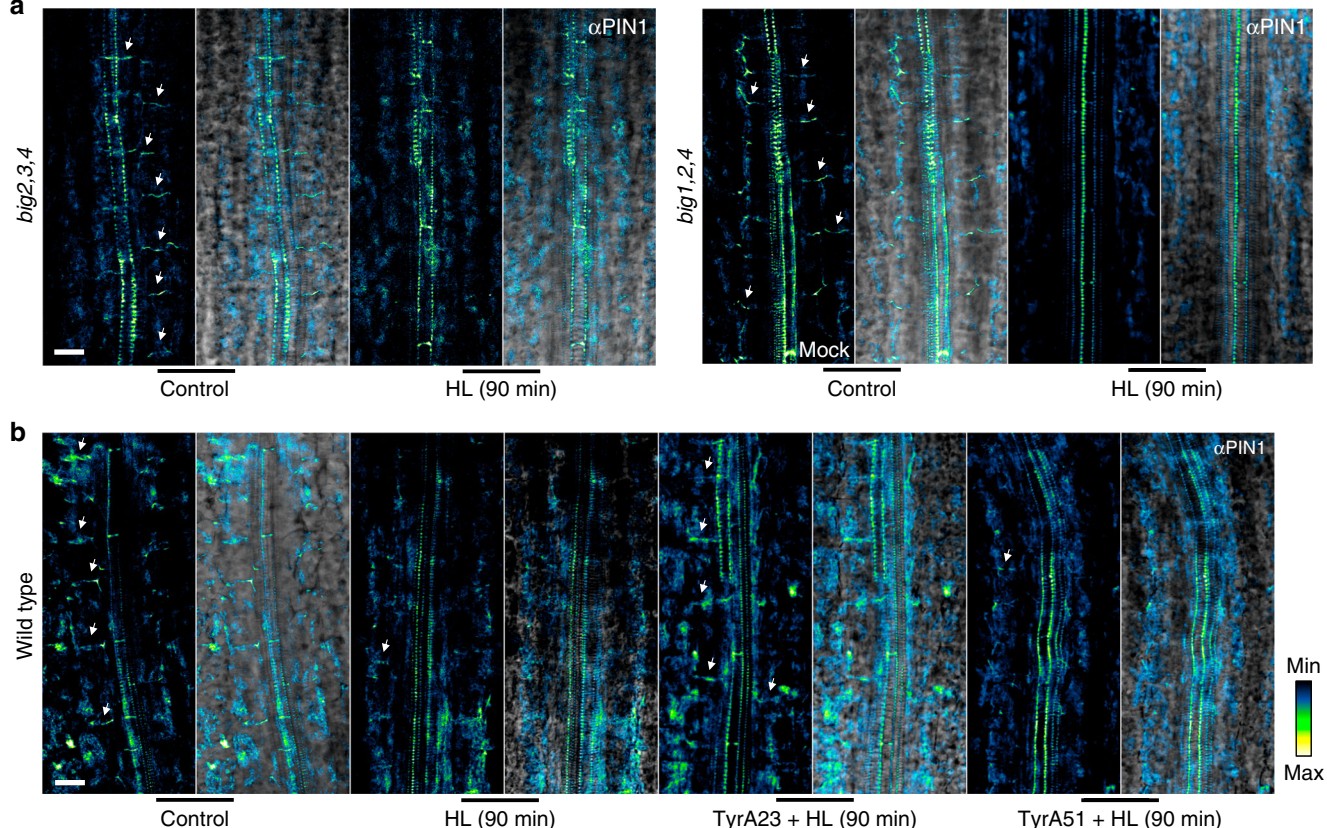

**Fig. 6** High light induces clathrin-mediated PIN1 endocytosis. **a** Representative dark field (left) and bright field (right) images of PIN1 immunolocalization in 7-day old hypocotyls of *big2,3,4* and *big1,2,4* seedlings before (control) and after 90 min of HL resulting in reduced PIN1 fluorescence intensity at the PM. **b** PIN1 immunolocalization dark field (left) and merged image with bright field (right) in 7-day old untreated (control) and 90 min after HL treatment, HL/Tyr23, and HL/TyrA51, showing reduced PIN1 abundance at the PM after HL and HL/TyrA51 treatment but not with HL/TyrA23. The color-coded bar displays the PIN1 fluorescence intensity. Representative images of two independent experiments, $n \geq 8$ biological replicates. Scale bar: 20 μm

stress-response circuitry of selected genes is well established[54–57]. Indeed uncovering MEcPP mode of action in transcriptional regulation of auxin biosynthetic genes is central for gaining insight into the molecular basis of the interplay between these two ancient and essential signaling pathways.

Moreover, we demonstrate decreased auxin flux through reduced abundance of PIN1 in constitutive and inducible MEcPP-producing lines as well as in plants exposed to HL, a physiologically relevant stress that result in accumulation of MEcPP. This finding confirms the notion of HL-mediated initiation of retrograde signal(s) that lead to acclimatory responses[58], and further identifies MEcPP as the stress-specific retrograde signal that accumulates as the result of HL treatment, mediating the observed adaptive responses. The reduced PIN1 protein abundance in the absence of any detectable changes in the polarity of this transporter, strongly suggests that MEcPP path of action is in reduction of PIN1 protein abundance and not via a change in the direction of auxin transport. In addition, targeted pharmacological interference with clathrin-dependent endocytosis of PIN1 identified post-transcriptional mode of MEcPP-mediated action through regulation of PIN1 abundance at the plasma membrane.

In summary, our results provide insight into the dynamic nature of plastidial regulation of auxin homeostasis and offer molecular evidence for the dual regulatory action of the stress-specific retrograde signaling metabolite, MEcPP, in modulating auxin and PIN1 abundance levels by transcriptional and post-translational regulatory inputs. A simplified schematic model depicts the dual path of MEcPP action constituting plastidial

operational mode of function in adjusting growth and reallocation of resources to adaptive responses (Supplementary Fig. 10).

This work provides a coherent picture of how the interplay between MEcPP and auxin homeostasis provides plants with the plasticity necessary to exert a refined control over the continuous environmental variables. In addition, and of particular importance is the concept of stress-induced plastidial retrograde metabolite based signaling responsible for regulation of growth, thereby shifting the paradigm of the role of the plastid in plant adaptive responsiveness from that of a secondary player to that of an essential primary component. Furthermore, the interconnection between auxin and MEcPP, two ancient and essential signaling metabolites, provides a fresh perspective on the evolution of adaptive mechanisms.

## Methods

**Plant growth and treatments**. All experiments were conducted on seedlings grown in long day (LD; 16-h light/8-h dark cycles), at ~22 °C on 1/2 × MS media. The seedlings used were *ceh1* and Parent (the *ceh1* background prior to EMS mutagenesis)[39], *HDSi* lines (see below), *DR5*:GFP lines that were kindly provided by Mark Estelle, (PIN2:PIN2-GFP, PIN3:PIN3-GFP, NPSN12-YFP[49,59,60]), and DII-VENUS[19], BIG mutant lines[26].

MEcPP treatment was performed as previously described with slight modifications. Specifically, 7-old seedlings were treated with MEcPP prepared in 1/2 MS to the final concentration of 100 μM. The mock experiments were conducted with 1/2 MS.

Tyrphostin A23 and tyrphostin A51 treatments were conducted on 7-day-old seedlings grown under LD condition. Seedlings were treated for 60 min with DMSO (control), or 100 μM tyrphostin A23 and 100 μM tyrphostin A51. The treated seedlings were subsequently exposed to HL (800 μmol m$^{-2}$sec$^{-1}$) for 90 min prior to immunolocalization studies.

**Hypocotyl length measurement**. 7-day-old seedlings were scanned using the Epson scanner, and ImageJ (from http://rsb.info.nih.gov/ij/) was used to measure the hypocotyl length.

**Quantification of gene expression**. Real-time quantitative PCR and data normalization were performed as described[43,61,62], using Quantprime. Each experiment was performed on three biological replicates each with three technical replicates, using the primer sequences shown in Supplementary Table 1.

**High light treatment**. Plate grown 7-day old seedlings were exposed to HL (800 μmol m$^{-2}$s$^{-1}$) for 30 min, 60 min and 90 min, at a controlled temperature maintained at 22 °C. At each of the indicated time points, seedlings were collected for MEcPP measurement, RT-qPCR, immunolocalization and confocal imaging.

**Immunolocalization of PIN1**. The immunolocalization analyses were carried out according to the described method[63] with some modifications. Specifically, we used 7-day-old seedlings that were fixed in 4% paraformaldehyde (PFA in MTSB) for 1 h at room temperature (RT). Upon removal of fixative the seedlings were initially washed with MTSB buffer and finally rinsed with H$_2$O. Seedlings were them placed on adhesive coated slides till dry before covering them with a coverslip chamber and subsequent addition of 200 μl 2% Driselase Basidiomycetes sp (Sigma), followed by application of vacuum for 3 min, and 30 min incubation at RT for 30 min. Next seedlings were washed 5 times with 200 μl MTSB followed by addition of 200 μl of 10% DMSO + 3% Igepal (Sigma) and incubation at RT for 1 h. Seedlings were then thoroughly washed several times with 200 μl MTSB, followed by addition of 200 μl 5% BSA (Sigma) and application of vacuum, and subsequent incubation at RT for 1 h. Next 200 μl of PIN1 antibody (1:50) in 5% BSA was added to the chamber, followed by incubation at 4 °C overnight, and subsequent incubation at 37 °C for 2 h. The seedlings were then washed with 200 μl MTSB before application of 200 μl of FITC anti-mouse (1:400) secondary antibody (KPL, 02-18-06) in 5% BSA, and incubation at 37 °C for 2 h. Seedlings were subsequently rinsed several times with 200 μl MTSB. Next MTSB was removed and seedlings were incubated in equilibration buffer (Invitrogen Antifade Kit) for 10 min at RT. Lastly, antifade reagent (Invitrogen Antifade Kit) was added to the chamber and finally sealed the slides with nail polish for confocal imaging.

**MEcPP and auxin measurements**. MEcPP extraction and quantification were performed as previously described[39,43]. Briefly, samples were analyzed using a Dionex Ultimate 3000 binary RSLC system coupled to Thermo Q-Exactive Focus mass spectrometer with a heated electro spray ionization source. Plant samples and standards were separated using an Accucore-150-Amide-HILIC column (150 × 2.1 mm; particle size 2.6 μM; Thermo Scientific 16726-152130) with a guard column containing the same column matrix (Thermo Scientific 852-00; 16726-012105). The separation was conducted in isocratic conditions using 60% acetonitrile with 0.1% formic acid and 40% 50 mM ammonium formate buffer pH 4.5. Flow rate was kept at 150 μL/min and the volume injected was of 5 μL. The column was kept at room temperature. Mass spectra were acquired in negative ion mode under the following parameters: spray voltage, 4.5 KV; sheath gas flow rate of 15 and capillary temperature of 275 °C. Samples were quantified using an external standard curve of MEcPP (Echelon, I-M054) with concentrations of 200, 100, 75, 60, 45, 36, 27, 13.5, 6.75 μM and final quantification were normalized to starting fresh weight.

 IAA extraction was performed as previously described by some modifications[64,65]. Specifically, plant materials were ground twice using bead beater (Mini-Beadbeater; Biospecs Products) under cryogenic condition, followed by extraction in isopropanol: water: HCl (2:1:2), and subsequent sonication and centrifugation for 5 min at 21,000 × g at 4 °C. Supernatants were collected and the pellets were re-extracted. Supernatants from both extractions were combined and filtered through a 0.22 μm PTFE filter (Waters, Milford, MA, USA). The samples were freeze dried (FreeZone Plus 4.5 Liter Cascade, LABCONCO, MO) and re-dissolved in of acetonitrile: water (80:20) and centrifuged (21,000 × g for 5 min) and transferred to liquid chromatography (LC) vials for injection.

**Liquid chromatography and mass spectrometry**. Samples were analyzed using a Dionex Ultimate 3000 binary RSLC system coupled to Thermo Q-Exactive Focus mass spectrometer with a heated electro spray ionization source. Samples separation and gradient elution was by acetonitrile containing 0.1 % formic acid (A) and water containing 0.1 % formic acid (B) by a gradient profile (t(min), %A, %B): (0, 5, 95), (20, 95, 5), (25, 95, 5), (25.01, 5, 95), (35, 5, 95), using a Acclaim$^{TM}$ RSLC 120 C18 column (100 × 2.1 mm, particle size 2.2 μM; Thermo Scientific 068982). The flow rate was maintained at 200 μl/min, and at 35 °C. Mass spectra in positive mode were acquired under the following conditions: spray voltage, 4.50 KV; sheath gas flow rate 45, auxiliary gas flow rate 20, sweep gas flow rate 2, capillary temperature of 250 °C, S-lens RF level 50 and auxiliary gas heater temperature 250 °C. For relative quantitation, peak area for each compound (MS2; Thermo Trace Finder Software) was normalized to weight.

**Root zone analysis using iRoCS pipeline**. Five days old seedlings of P and ceh1 were transferred to liquid ½ MS medium for 5 h. Thereafter 10 μm 5-ethynyl-2′-

deoxyuridine (EdU) was added for 90 min. Seedlings were fixed in 4% formaldehyde for 60 min in microtubule stabilization buffer[45]. EdU was detected according to the manufacturer's manual with modifications previously described[45]. After EdU detection, roots were washed twice with distilled water for 10 min, incubated in 200 μg L$^{-1}$ 4′,6-diamidino-2-phenylindole (DAPI) for 20 min, washed again with distilled water and mounted on slides with a 120 μm spacer using mounting medium (DAPI GOLD reagent; Thermo Fisher Scientific Inc., Waltham, MA, USA). DAPI/EdU-stained samples were recorded using a confocal laser scanning microscope (ZEISS LSM 510 META NLO) with a LD LCI-Plan-Apochromat 25 × /0.8 DIC Imm Korr objective. For the DAPI excitation, a 740 nm Chameleon laser was used and emission was detected with a band pass filter (BP 390–465 IR); EdU excitation was at 488 nm and emission was detected with a band pass filter (BP 500–550 IR). Serial optical sections were reconstituted into 3D image stacks to a depth of 100 μm with in-plane (x–y) voxel extents of 0.15 and 0.9 μm section spacing (z). Two or three overlapping images (tiles) were recorded for each root. Images were converted to hdf5 format and then stitched. Representative roots for each treatment were chosen for annotation. Nuclei, mitosis events and DNA replication events were annotated using the iRoCS Toolbox[44]. All analyses and graphical presentations were performed as described[66].

**Microscopy**. Confocal fluorescence imaging was performed using a Leica TCS SP5 confocal microscope (Leica Microsystems) or a Zeiss LSM 880 upright (Zeiss international). The manufacturer's default settings were used for imaging GFP-, VENUS-tagged proteins and FITC fluorophore. Fluorescence signal for DR5-GFP and DII-VENUS[19] was detected with 10× water objective, PIN1-GFP, PIN1-FITC, PIN3-GFP, NPSN12-YFP, and PIN2:GFP[59] were detected with 40× objective. Signal intensity quantifications were performed using the ImageJ software (imagej.nih.gov/ij/).

**Western blot analyses**. Protein extraction was performed on 7-day old P and ceh1 seedlings grown on a 1/2 × MS media. Tissue was frozen upon collection and grind in liquid nitrogen using protein extraction buffer (50 mM Tris-HCl, pH 8, 10 mM EDTA, 2 mM EGTA, 0.01% SDS, 1 mM DTT, 10 μM/ml Protease Inhibitor Cocktails (Sigma)). The extract was subsequently centrifuged at 10,000 RPM for 15 min at 4 °C. The supernatant was collected and protein concentration was measured using Thermo Scientific Pierce Micro BCA Assay according to manufacturer instructions. Protein concentration of samples were adjusted to 2 or 3 μg/μL in 5 × Laemmli Buffer, heated at 56 °C and subsequently separated on 10% SDS-PAGE gel and transferred onto nitrocellulose membranes. Blots were probed with anti-PIN1 monoclonal antibody (1:100)[63] primary antibody and secondary anti-mouse-Horseradish peroxidase (HRP) (KPL, catalog no. 074-1806) (1:3000) and chemi-luminescent reaction was performed using and Pierce ECL Western Blotting Substrate. Working solutions of the substrates were prepared according to the manufacturers' instructions and added to the membranes. The membranes were placed in plastic sheet protectors. Each membrane was exposed to X-ray films and developed. The uncropped scans of the blots are shown in Supplementary Figs. 1d and 6b. The anti-PIN1 monoclonal antibody was produced in Klaus Palme's laboratory.

**Inducible RNAi line of _HDS_**. Homozygous Dexamethasone (DEX) inducible _HDSi_ lines were generated by transforming plants with RNAi _pOpOff_ vector construct[67] harboring the _HDS_ cloned with primer sequences shown in Supplementary Table 1. The seedlings were grown under LD condition for 7 days before treatment with 30 μM of DEX, followed by sample collection at 0, 48, and 72 h post induction.

**Statistical analysis**. All of the experiments were performed with at least three biological replicates each with three technical replicates. Data are mean ± standard deviation (SD). These analyses were carried out via a two-tailed Student's t tests or R program with a significance of $P < 0.05$.

**Data availability**. The authors declare that all data supporting the findings of this study are available within the manuscript and its supplementary files.

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

## Acknowledgements

The authors would like to thank Glenn Hicks for critical discussions. We also express our thanks to Drs. Jared Shaw and Edward Balmond for providing the chemically synthesized MEcPP. This work was supported by Bundesministerium für Bildung und Forschung (BMBF SYSBRA, SYSTEC, Microsystems), DLR and German Research Foundation (DFG SFB746, INST 39/839,840,841), and by the National Science Foundation (NSF) IOS-1036491, NSF IOS-1352478, and National Institutes of Health (NIH) R01GM107311 to K.D.

## Author contributions

J.J., C.R.F., and K.D. designed the study, J.J., C.R.-F., J.-Z.W., A.d.S., H.K., T.P., H.L., F.A. D. performed the experiments, K.P. provided enabling reagents, and K.D. wrote the manuscript.

## Additional information

**Competing interests:** The authors declare no competing interests.

