## [Peer Review File · Nature Communications]

Reviewers' comments:

Reviewer #1 (Remarks to the Author):

Reviewed by Dior Kelley 1/9/18

Summary:

- In this manuscript Jiang & Furlan et. al., provide interesting new data on complex plant cellular communication networks, which would be of interest to the auxin community and plant biologists in general.
- Specifically, the authors present exciting evidence for how plastid-to-nucleus retrograde signaling events in young vegetative tissues (hypocotyl and primary root) can simultaneously influence auxin pathways (biosynthesis and transport). The data presented here provide a possible direct molecular mechanism to describe how "auxin-based signaling as secondary components involved in the response cascades following a plastidial signal" (Gläber et. al., Mol Plant 2014) and a model for how high light can alter auxin accumulation and auxin responsive gene expression (Gläber et. al., Mol Plant 2014; Gordon et. al., Front Plant Sci 2013). However, neither of these papers were cited.
- Additionally, the regulation of endocytosis via retrograde signaling conveyed by the plastidial metabolite MEcPP is quite interesting and is a very novel finding.

Main comments:

- In the introduction, there needs to be a transition between discussing auxin pathways and exploitation of the *ceh1* mutant; this is a novel connection of two discrete metabolite pathways that have not previously been linked. Perhaps mentioning of earlier work suggesting links between plastidial communication, auxin response and accumulation (for example, Gläber et. al., Mol Plant 2014; Gordon et. al., Front Plant Sci 2013) could be included as a logical progression? Or some further elucidation by the authors as to how they made the connection between MEcPP and auxin signaling.
- Using the *ceh1* mutant and pharmacological experiments the authors show that *ceh1* plants exhibit shorter hypocotyls compared to wild-type, have reduced endogenous levels of IAA, reduced YUC3,5 expression, reduced PIN1,2,3 protein levels and reduced auxin responsiveness. Thus, increased levels of MEcPP reduces IAA biosynthesis, which in turn dampens transcriptional auxin responses (measured via DR5:GFP and DII:VENUS) and transport (measured via PIN localization and abundance). Are there stable or inducible overexpression HDS lines that could be examined to see if the phenotypes are consistent?
- Figure 2: Reduced cell division in *ceh1* roots is consistent with reports of auxin levels and/or transport being linked to cell division activity in the root, but the data shown here does not show that in this instance the process is auxin dependent per se. Since exogenous application of IAA can restore hypocotyl length in *ceh1* to a wild-type length (Figure 1B), can the same treatment restore cell division in the primary root (Figure 2)? If not, perhaps the authors can speculate on this.
- Figure 3: The "further reduction" of PIN1-GFP after 30 min in Figure 3d is not striking from the image shown; are there other more representative roots or is it just that PIN1 levels are dampened within a narrow region of the root tip, such as the meristem? The reduction of PIN1-GFP in the hypocotyl shown in Figure 3c is clearer than for the root (Fig 3d).
- Figure 4: builds on a previous study that shows retrograde signaling pathways impinge upon auxin response (Gläber et. al., Mol Plant 2014; <https://www.ncbi.nlm.nih.gov/pubmed/24719466>) and provides evidence that such a phenomenon is mediated by MEcPP. This paper should be cited here and again discussed further in the discussion section.
- Figure 5: high light induces clathrin mediated endocytosis of PIN1. Would it be possible to perform FM4-64 staining (or similar fluorescent dye) to co-image endocytic vesicles in these conditions? While the punctate structures are consistent with endocytic vesicles some sort of co-staining or marker for these organelles would provide stronger evidence.

- What does a *ceh1 yucca-1d* double mutant look like? If reduced YUCCA expression is responsible for the auxin-dependent short hypocotyl phenotype in *ceh1* plants then overexpression of YUCCA in the *ceh1* background should directly address this hypothesis.
- Finally, I'd like to see the authors expand the discussion to include a more detailed model for how MEcPP alters indole-3-acetic acid levels and PIN1 accumulation; for instance, (1) how might a small molecule such as MEcPP directly regulate YUC3 and YUC5 transcription? (2) how might MEcPP regulate endocytosis? And (3) how does MEcPP act to control protein abundance of specific proteins such as PIN1? For instance, does MEcPP regulate mRNA stability, translation, protein degradation or protein localization? Or does MEcPP act at the transcriptional level and protein abundance is well correlated with mRNA levels for the genes examined here (YUCCAs and PINs)? Some further detailed hypotheses could be presented here regarding the direct modes of molecular action of MEcPP need to be clarified here. The authors might consider adding a corresponding schematic figure illustrating these ideas to help the readers; this could be added to the supplemental file.

Minor comments:

1. Check that the font is the same throughout the manuscript; the introduction appears to have a couple different fonts (but perhaps that is a pdf conversion issue).
2. Check for grammar/spelling throughout; e.g. "hemostasis" should be "homeostasis"; "posttranscriptional" should be hyphenated "post-transcriptional".
3. Figure 1C – consider showing the percent of hypocotyl increase for each genotype rather than absolute length since *ceh1* hypocotyls are shorter than P hypocotyls.
4. Figure 4 legend: suggest editing the text from "High light reduces DR5 and PIN1 abundance" to "High light reduces DR5:GFP and PIN1 abundance".
5. Are the primers used for qPCR listed in the manuscript? I did not see a table in the methods or supplemental files but perhaps I missed it. These sequences should be added to the methods and/or included as a supplemental table.
6. Materials & Methods: "Auxin measurement as previously described". Specify which endogenous auxin form was measured (I believe it was IAA?).
7. Figure S4 c-e legend: state what part of the plant is being imaged.

Reviewer #2 (Remarks to the Author):

This manuscript describes a connection between the retrograde signaling molecule MEcPP and the plant hormone auxin. In this work, the authors show that the *ceh1* mutant, a mutant that hyperaccumulates MEcPP, displays decreased hypocotyl elongation, decreased auxin levels, and decreased expression of the YUC3 and YUC5 auxin-biosynthesis genes. Further, *ceh1* displays decreased PIN1 transcript and protein.

Making connections between chloroplast-generated MEcPP and auxin would certainly be interesting; however, the direct link between these is lacking for me. It appears that MEcPP affects all aspects of auxin biology (biosynthesis, transport, and signaling), but the molecular mechanism of this is unclear. Further, it is unclear whether these effects are from MEcPP itself or are a product of increasing the isoprenoids downstream of this metabolite.

A larger concern for me is that the data presented do not meet the standards in the field – in particular, all data collected was with a single allele, without a complementation line. Data collected under such circumstances has plagued the auxin field in recent years. It is possible that the observed effects on auxin are due to a background mutation, and may account for inconsistencies observed between the *ceh1* mutant and treatment with MEcPP.

Specific comments:

- 1- Either a second *ceh1* allele or a *ceh1* complementation line need to be characterized to verify data.
- 2- Were the HL experiments (Fig. 5) done in hypocotyl or root tissues? If roots, would you expect this to be physiologically relevant? Is MEcPP transported from aerial tissues to the root?
- 3- In line with the above comment, the authors switch between using hypocotyl and root tissues in their assays. It is not clear to me how one might expect MEcPP effects in the root, which was not well-explained in the text.
- 4- I don't understand what is being described in Fig. 2B. Is the distance reported the distance of EdU-positive cells from the QC? This is not well described in the text or figure legend.
- 5- The authors suggest that the decreased hypocotyl length observed in *ceh1* is caused by decreased auxin levels as a result of deficient YUC3 and YUC5 transcript levels. However, the *yucQ* mutant, deficient in YUC3,5,7,8,9, displays wild type hypocotyl length (Chen 2014). This needs to be explained or the wording of this altered.

Minor comments:

Fig. S5a- Root cell sizes appear different between mock and MEcPP-treated seedlings. This seems strange / unusual.

To me, using the intensity scale in the microscopy images make them difficult to see.

Reviewer #3 (Remarks to the Author):

This paper addresses the interesting question of how stress signals such as accumulation of MEcPP slow down plant growth. Does MEcPP function as an active growth inhibiting signal or by interfering with growth promoting pathways? The authors present a number of lines of evidence that MEcPP acts by interfering with the growth promoting pathway of auxin, specifically by reducing auxin levels and PIN1 abundance. They argue that this is a specific interaction between MEcPP and auxin, with no involvement of salicylic acid. Altogether, their data suggest one mechanism by which growth-inhibiting signals like MEcPP function is by interfering with growth promoting pathways, such as auxin. The authors argue that the novelty of their results is that they have connected retrograde signaling from plastids to growth responses.

This paper is missing a few key experiments and lacks a thorough Discussion.

1. Other labs have shown that growth promoting or inhibitory signals originate in the epidermis. For instance, small secreted peptides, called RALFs, may act to inhibit plant growth by repressing the inherent tendency of the inner tissues to elongate. RALFs suppress the growth-promoting hormones, GAs and BRs, while promoting growth-inhibiting hormones, ethylene and JA. Are the authors sure that MEcPP acts exclusively through auxin?

2. What cell types/plastid types accumulate MEcPP?

An RNASeq timecourse during the first hour after treatment with MEcPP should begin to answer both of these questions. Question #2 would be better answered with an RNASeq on an epidermal peel after treatment with MEcPP; however, this experiment may have to be done using Brassica.

Finally, this paper would be greatly improved by a thorough Discussion of the numerous papers on plant growth.

Reviewer #1

We are particularly grateful to the insightful comments of this reviewer who has clearly identified some conceptual oversights. Thus, we have fully complied and included the following data to address the reviewer's concerns and comments.

1. *The data presented here provide a possible direct molecular mechanism to describe how “auxin-based signaling as secondary components involved in the response cascades following a patidial signal” (Gläßer et. al., Mol Plant 2014) and a model for how high light can alter auxin accumulation and auxin responsive gene expression (Gläßer et. al., Mol Plant 2014; Gordon et. al., Front Plant Sci 2013). However, neither of these papers were cited.*

R1. This is indeed an oversight that is now corrected. These two papers are included and discussed within the context of retrograde signaling.

2. *In the introduction, there needs to be a transition between discussing auxin pathways and exploitation of the *ceh1* mutant; this is a novel connection of two discrete metabolite pathways that have not previously been linked.*

R2. We have addressed this shortcoming and have provided the much needed conceptual transition between the two pathways.

3. *Are there stable or inducible overexpression HDS lines that could be examined to see if the phenotypes are consistent?*

R3. Per reviewer's suggestion we have employed DEX-inducible MEcPP-producing line (*HDSi*) and confirmed the consistency of the phenotypes observed, namely reduced IAA and PIN1 abundance, upon DEX induction and by extension MEcPP accumulation. This results confirms the earlier data obtained from constitutively high MEcPP containing *ceh1* mutant line (Figs 3a-c).

In addition, we have confirmed the above mentioned phenotypes using fosmidomycin (FSM), an inhibitor that arrest the flux through the MEP pathway, and reversing the aforementioned phenotypes (Figs 2a-b, &S6).

4. *Since exogenous application of IAA can restore hypocotyl length in *ceh1* to a wild-type length (Figure 1B), can the same treatment restore cell division in the primary root (Figure 2)? If not, perhaps the authors can speculate on this.*

R4. We have performed these experiment and show differential inhibition of root growth in *ceh1* versus P plant in response to auxin treatment. These data are shown (Fig S4), and the results are discussed. We assume that higher auxin levels may be absorbed at the roots than at hypocotyls, and as such the concentration necessary to restore the hypocotyl growth might result in inhibition of root growth.

5. *Figure 3: The “further reduction” of PIN1-GFP after 30 min in Figure 3d is not striking from the image shown; are there other more representative roots or is it just that PIN1 levels are dampened within a narrow region of the root tip, such as the meristem? The reduction of PIN1-GFP in the hypocotyl shown in Figure 3c is clearer than for the root (Fig 3d).*

R5. We agree, and thus we have changed the color coding to better display the differences.

6. *Fig4 builds on a previous study that shows retrograde signaling pathways impinge upon auxin response (Gläßer et. al., Mol Plant 2014; <https://www.ncbi.nlm.nih.gov/pubmed/24719466>) and provides evidence that such a phenomenon is mediated by MEcPP. This paper should be cited here and again discussed further in the discussion section.*

R6. We have fully complied by citing and discussing the Gläßer et. al manuscript.

7. *Figure 5: high light induces clathrin mediated endocytosis of PIN1. Would it be possible to perform FM4-64 staining (or similar fluorescent dye) to co-image endocytic vesicles in these conditions? While the punctate structures are consistent with endocytic vesicles some sort of co-staining or marker for these organelles would provide stronger evidence.*

R7. We are in full agreement with the reviewer, but due to limitations such as availability of antibodies we were unable to perform the suggested experiment for the current manuscript.

8. *What does a *ceh1 yucca-1d* double mutant look like? If reduced YUCCA expression is responsible for the auxin-dependent short hypocotyl phenotype in *ceh1* plants then overexpression of YUCCA in the *ceh1* background should directly address this hypothesis.*

R8. Both these mutants are very stressed so it is difficult to conclusively arrive at an answer when there are so many other pleiotropic phenotypes.

9. *Finally, I'd like to see the authors expand the discussion to include a more detailed model for how MEcPP alters indole-3-acetic acid levels and PIN1 accumulation; for instance, (1) how might a small molecule such as MEcPP directly regulate YUC3 and YUC5 transcription?*

The authors might consider adding a corresponding schematic figure illustrating these ideas to help the readers; this could be added to the supplemental file.

R9. We have now included a model as suggested in supplementary data (Fig. S10) and discussed the potential mode of MEcPP action.

Minor comments:

1. *Check that the font is the same throughout the manuscript; the introduction appears to have a couple different fonts (but perhaps that is a pdf conversion issue).*

r1. We have checked the font, it was uniform throughout the manuscript, and perhaps the inconsistency occurred through conversion to PDF?

2. *Check for grammar/spelling throughout; e.g. "hemostasis" should be "homeostasis"; "posttranscriptional" should be hyphenated "post-transcriptional".*

r2. We apologize for the oversight, the corrections are done.

3. *Figure 1C – consider showing the percent of hypocotyl increase for each genotype rather than absolute length since *ceh1* hypocotyls are shorter than *P* hypocotyls.*

r3. We agree, but in our experience many reviewers demand the absolute value, thus we have displayed it as such.

4. *Figure 4 legend: suggest editing the text from “High light reduces DR5 and PIN1 abundance” to “High light reduces DR5:GFP and PIN1 abundance”.*

r4. We have complied.

5. *Are the primers used for qPCR listed in the manuscript? I did not see a table in the methods or supplemental files but perhaps I missed it. These sequences should be added to the methods and/or included as a supplemental table.*

r5. The primer sequences are included in table 1.

6. *Materials & Methods: “Auxin measurement as previously described”. Specify which endogenous auxin form was measured (I believe it was IAA?).*

r6. We have included the entire method, and the measurement was indeed for IAA.

7. *Figure S4 c-e legend: state what part of the plant is being imaged.*

r7. The image is from hypocotyl, and information is included in the legend.

Reviewer#2

1. *It is unclear whether these effects are from MEcPP itself or are a product of increasing the isoprenoids downstream of this metabolite. A larger concern for me is that the data presented do not meet the standards in the field – in particular, all data collected was with a single allele, without a complementation line. Data collected under such circumstances has plagued the auxin field in recent years. It is possible that the observed effects on auxin are due to a background mutation, and may account for inconsistencies observed between the *ceh1* mutant and treatment with MEcPP.*

R1. We are in full agreement with the reviewer, therefore we have addressed the reviewer's concern by repeating the experiment using two other approaches, namely inducible *HDSi* line as well as treatment of plants with fosmidomycin, as indicated in R3 to reviewer#1.

With these additional experiments we have the consistent results obtained from parallel approaches using:

1. Constitutively high MEcPP producing plants (*ceh1*)
2. Inducible MEcPP producing lines (*HDSi* line)
3. Inhibition of the MEP pathway by FSM
4. External application of MEcPP

We believe that the combination should sufficiently address the reviewer's concerns.

2. *Were the HL experiments (Fig. 5) done in hypocotyl or root tissues? If roots, would you expect this to be physiologically relevant? Is MEcPP transported from aerial tissues to the root?*

R2. The plate grown seedlings were exposed to high light, and the analyses were exclusively performed on hypocotyl.

3. *In line with the above comment, the authors switch between using hypocotyl and root tissues in their assays. It is not clear to me how one might expect MEcPP effects in the root, which was not well-explained in the text.*

R3. Root data were only from exogenously MEcPP treated plants to demonstrate the effectiveness of MEcPP even when applied the exogenously.

4- *I don't understand what is being described in Fig. 2B. Is the distance reported the distance of EdU-positive cells from the QC? This is not well described in the text or figure legend.*

R4. We have modified the presentation of the figures and rewritten the figure legend for clarity.

5- *The authors suggest that the decreased hypocotyl length observed in *ceh1* is caused by decreased auxin levels as a result of deficient *YUC3* and *YUC5* transcript levels. However, the *yucQ* mutant, deficient in *YUC3,5,7,8,9*, displays wild type hypocotyl length (Chen 2014). This needs to be explained or the wording of this altered.*

R5. We agree with the reviewer, hence the result of the *yucQ* study (Chen 2013) is included and discussed accordingly.

Minor comments:

Fig. S5a- Root cell sizes appear different between mock and MEcPP-treated seedlings. This seems strange / unusual.

r1. This is due to naturally occurring variations between root sizes.

Reviewer#3

1. *Other labs have shown that growth promoting or inhibitory signals originate in the epidermis. For instance, small secreted peptides, called RALFs, may act to inhibit plant growth by repressing the inherent tendency of the inner tissues to elongate. RALFs suppress the growth-promoting hormones, GAs and BRs, while promoting growth-inhibiting hormones, ethylene and JA. Are the authors sure that MEcPP acts exclusively through auxin?*

R1. We agree with the reviewer that there are instances when the signal originates in the epidermis. However, the signal identified here is a plastidial signal present in all cells containing chloroplast. Indeed, we hope that the newly performed experiments (see the R3 response to reviewer #1 and R1 to reviewer#2) have confirmed the function of MEcPP, the plastidial retrograde signal, in altering auxin homeostasis.

2. *What cell types/plastid types accumulate MEcPP?*

R2. All chloroplast containing cells.

3. *An RNASeq timecourse during the first hour after treatment with MEcPP should begin to answer both of these questions. Question #2 would be better answered with an RNASeq on an epidermal peel after treatment with MEcPP; however, this experiment may have to be done using Brassica.*

R3: Unfortunately due to the high price of MEcPP (\$500/100 µg) we are unable to perform RNA seq after MEcPP treatment, and also conducting these experiments in Brassica is not within the current scope of this work. Lastly, MEcPP is produced in all chloroplast containing cells and as such using epidermal peel might not be as informative.

4. *Finally, this paper would be greatly improved by a thorough Discussion of the numerous papers on plant growth.*

R4. We agree with the reviewer and as such we have extended the discussion section to fully capture the importance of the data

REVIEWERS' COMMENTS:

Reviewer #1 (Remarks to the Author):

In this revised manuscript, the authors have addressed all my comments sufficiently. The additional experiments and textual changes have greatly added clarification and support for their conclusions.

Reviewer #2 (Remarks to the Author):

The authors have adequately responded to my previous concerns.

Comments to the Reviewers

We greatly appreciate your time and input.